# When Encoders Should Stay Simple: An Empirical Analysis of Architectures for Variational Autoencoders

## Abstract

Variational Autoencoders (VAEs) offer efficient training and inference without the reliance on computationally expensive Monte Carlo Markov Chain (MCMC) methods. Despite their foundational importance, the architectural choices for encoders and decoders in VAEs remain underexplored, particularly their impact on the learned latent representations and generative quality. This study investigates the influence of encoder and decoder architectures by systematically varying their configurations using dense and convolutional network-based models. Experiments were conducted across different latent space sizes to assess the models' compressive capacities and performance on reconstructive and generative losses.

The results reveal that small dense networks are more effective for encoding, while decoding benefits from architectures with structural processing capabilities, such as convolutional networks with multiple blocks. Dimensionally bigger latent space compression levels degrade representation quality but maintain separability at moderate compression levels. Notably, models with non-zero Kullback-Leibler Divergence (KLD) loss outperform collapsed latent space models, emphasizing the importance of balancing reconstruction and generative regularization. These findings provide insights into the architectural considerations necessary for designing efficient VAEs and improving their generative and representational capabilities.

## 1 Introduction

Probabilistic generative models aim to discover meaningful data representations while modeling complex probabilistic distributions. These models not only learn the underlying structure of data but also enable the generation of synthetic samples. However, general probabilistic generative models often suffer from intractable likelihoods, necessitating computationally expensive training algorithms that rely on Monte Carlo Markov Chain (MCMC) sampling methods.

Deep Boltzmann Machines (DBMs) Salakhutdinov & Hinton (2009) represent an early solution integrating deep learning architectures with generative modeling techniques. By introducing multiple layers, DBMs expanded the capacity of generative models to handle complex data distributions. Despite this, their training process is inefficient due to the reliance on MCMC sampling for estimating the variational lower bound. Methods like the Sleep-Wake algorithm for posterior approximation also suffer from similar inefficiencies.

In contrast, Variational Autoencoders (VAEs) introduced a more efficient approach by enabling online training of both generative and recognition modules without the need for expensive MCMC methods. VAEs utilize gradient-based parameter updates, allowing the integration of deep learning architectures without significant computational restrictions. This innovation provided a practical and scalable solution for training generative models Kingma & Welling (2022).

The amortized variational approach proposed by VAEs has led to significant advances in deep generative modeling. However, in terms of generation quality, VAEs have been surpassed by Generative Adversarial Networks (GANs) Goodfellow et al. (2014) and subsequent models. GANs are capable of producing sharper and more realistic images compared to VAEs, which are often criticized for

generating blurry results. This discrepancy is primarily attributed to the mismatch between the true posterior and the approximate posterior learned by VAEs.

Several factors contribute to the limitations of VAEs in generating high-quality samples:

- **Simplistic Posterior Assumptions**: The Gaussian posterior assumption in VAEs may lead to overly simplistic representations and averaged outputs, underestimating the true variance of the data. Works solving this matter are Luhman & Luhman (2021) and Kingma et al. (2016).
- **Inference Suboptimality**: Challenges in optimization and loss formulation can limit the model's ability to accurately infer the latent structure. This problem is addressed by works Higgins et al. (2017) and Tomczak & Welling (2018).
- **Posterior Collapse**: Issues such as posterior collapse, where the latent variables fail to carry meaningful information, are particularly pronounced in architectures like PixelVAEs Vahdat & Kautz (2020).

To address these challenges, several solutions have been proposed, including the development of tighter bounds, improved data modeling techniques, and more sophisticated priors. Comprehensive reviews of these approaches can be found in Vahdat & Kautz (2020) and Bond-Taylor et al. (2022). Most strategies focus on improving the approximate posterior by modifying the latent space's probabilistic structure or reformulating the loss function.

Interestingly, recent studies suggest that the architecture of the encoder and decoder plays a significant role in shaping the latent structure of VAEs Vahdat & Kautz (2020). This work explores the influence of encoder and decoder architectures on the performance of VAEs in a simplified setting, deliberately isolating other methods related to probabilistic inference. By doing so, we aim to provide insights into how architectural choices impact the quality of the learned latent representations and the overall model performance.

## 2 BACKGROUND

### 2.1 VARIATIONAL AUTOENCODER

The Variational Autoencoder (VAE) framework addresses the intractability of traditional generative modeling by avoiding multiple steps Monte Carlo Markov Chain (MCMC) estimation methods. This results in a more efficient training process, particularly for large datasets.

#### 2.1.1 GENERAL ARCHITECTURE

The VAE framework comprises two main components to perform generative modeling of data given a prior distribution $p_{\theta^*}(\mathbf{z})$:

- The *recognition model*, $q_\phi(\mathbf{z}|\mathbf{x})$, also referred to as the inference model or probabilistic encoder, aims to approximate the true posterior $p_\theta(\mathbf{z}|\mathbf{x})$.
- The *generative model*, $p_\theta(\mathbf{x}|\mathbf{z})$, also known as the probabilistic decoder, generates data conditioned on the latent variables $\mathbf{z}$.

Using the marginal log-likelihood formulation in generative modeling by mean of variational approximation, the Evidence Lower Bound (ELBO) is derived to optimize the generative parameters. The ELBO is expressed as:

$$\mathcal{L}(\boldsymbol{\theta}, \boldsymbol{\phi}; \mathbf{x}^{(i)}) = -\mathbb{D}_{\text{KL}}(q_\phi(\mathbf{z}|\mathbf{x}^{(i)})||p_\theta(\mathbf{z})) \\ + \mathbb{E}_{q_\phi(\mathbf{z}|\mathbf{x}^{(i)})}[-\log p_\theta(\mathbf{x}^{(i)}|\mathbf{z})]. \tag{1}$$

This expression can be interpreted as the sum of two terms:

- The first term regularizes the latent space inferred by the model, encouraging it to follow the prior distribution $p_{\theta^*}(\mathbf{z})$.
- The second term ensures that the generative model reconstructs the input data $\mathbf{x}$ using the latent variable $\mathbf{z}$, represented as a probabilistic distribution.

### 2.1.2 REPARAMETERIZATION TRICK

To overcome the inefficiencies of MCMC sampling for estimating expectations, the original VAE work introduced the Stochastic Gradient Variational Bayes (SGVB) estimator. This approach employs a differentiable transformation $g_\phi(\boldsymbol{\epsilon}, \mathbf{x})$ to reparameterize the inference model $q_\phi(\mathbf{z}|\mathbf{x})$, allowing $\mathbf{z}$ to be approximated as:

$$\mathbf{z} = g_\phi(\boldsymbol{\epsilon}^{(i,l)}, \mathbf{x}^{(i)}), \quad \boldsymbol{\epsilon}^{(l)} \sim p(\boldsymbol{\epsilon}), \tag{2}$$

where $i$ represents the data points and $l$ represents the samples drawn from the reparameterization.

## 2.2 RELATED AND DERIVATIVE WORKS

### 2.2.1 DEEP GENERATIVE STOCHASTIC NETWORKS

A related generative modeling approach, the Deep Generative Stochastic Networks (DGSN) Bengio et al. (2014), trains deep architectures using backpropagation over structured inputs in a generative manner. Unlike VAEs, DGSN are considered implicit generative models, following definitions mentioned in Mohamed & Lakshminarayanan (2017), employing a fundamentally different approach. In DGSN, the encoder is a stochastic transformation that "destroys" the input, while the decoder reconstructs the input based on a parameterized probabilistic model.

An important insight from DGSN is that a high-capacity decoder can recover data even from an arbitrarily simple encoder. Conversely, recovering data from a highly complex encoder is significantly more challenging. Although DGSN differ in several ways from VAEs, they share the fundamental goal of reconstructing data from a stochastically defined distribution.

### 2.2.2 NVAE

The NVAE model Vahdat & Kautz (2020) improves the generative capacity of VAEs by architecture design. Most deep architectures used in recognition and generation tasks are heavily based on those successful in classification tasks. However, classification task and generative modeling serve fundamentally different purposes: classification aims to discard irrelevant information to achieve accurate predictions, while generative modeling seeks to retain as much relevant information as possible from the data.

NVAE emphasizes the importance of architectural choices in designing effective VAEs. It highlights that optimizing mutual information between the latent representation and the data plays a crucial role in maximizing the model's ability to generate high-quality samples. This work underscores the overlooked role of architectural design in generative modeling, a perspective we further explore in this study.

## 3 METHOD

To study the dynamics of encoder and decoder architectures, the general VAE framework will be applied with various combinations of encoder and decoder architectures. These architectures are constructed using basic deep learning building blocks for both modules, with their capacity progressively increased in each experiment. This approach enables the training of multiple iterations of deep learning models, allowing the exploration of optimal architecture combinations and providing valuable insights into their behavior. Additionally, the experiments will examine the model's compressing capacity by varying the size of the latent space.

The models will be characterized based on their optimization objectives and visually evaluated for reconstruction quality. Furthermore, the decoupling quality of the latent codes will be analyzed through projections of the latent space using Principal Component Analysis (PCA), which helps avoid overfitting the representation.

All experiments are be conducted on the MNIST dataset Deng (2012).

For the architectural design, it is important to note that standard architectures designed for classification tasks are not inherently suitable for generative modeling. As a result, the architectures employed in this study are built using basic structures. For convolutional networks (CNNs) Lecun

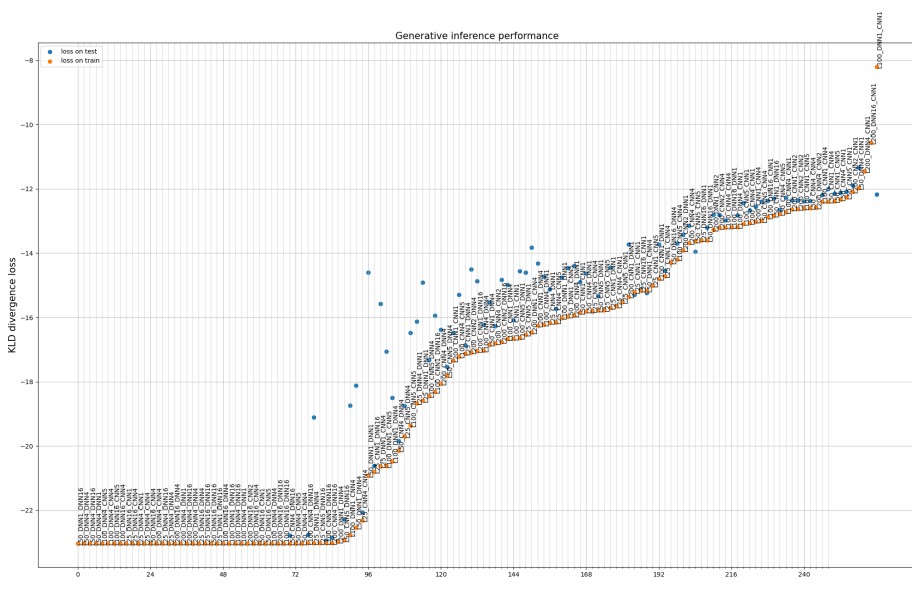

Figure 1: Performance of model combinations on the generative inference loss (log scale).Labels for each training follow the grammar L{latent space size}_{Encode architecture}{number of layers}_{Decoder architecture}{nnumber of layers}.

et al. (1998), the convolutional blocks consist of filters with a kernel size of $5 \times 5$ and a stride of 2, using LeakyReLU as the activation function. In case of convolutional decoding, decovolutional neural networks Zeiler et al. (2010) layers analogous to convolutioanl layers were used, for the rest of the work they will be refered as convolutional decoders. For dense networks (DNNs), the layers are implemented with matrix multiplication, biases, and LeakyReLU activation. The idea behind this design is returning to the basics, and explore posbible improvements from there.

## 4 RESULTS

### 4.1 GENERAL PERFORMANCE

Since the optimization objective of VAEs combines two distinct losses, the performance of each is analyzed separately. Figure 1 shows the generative loss performance of the models in a log scale, ordered by increasing loss values. It is evident that nearly half of the experiments result in collapsed latent spaces, this is latent space distributions being identical to a multivariate normal distribution. Such behavior generally implies that the model has not learned a meaningful representation of the dataset.

When ordered by reconstructive performance (Figure 2), a weak correlation is observed between models with collapsed or near-collapsed latent spaces and poor reconstructive performance.

Visual evaluation revealed that the top 25% of models have minimal reconstruction collapse. Among these models, a negative trend is observed in the generative inference loss when compared to reconstructive performance. This finding indicates that having a non-zero generative loss is generally beneficial for model performance (Figure 3).

### 4.2 ARCHITECTURE PERFORMANCE

The top 25% of models were further analyzed to identify the architectural configurations that performed best for encoding and decoding tasks. Figure 4 reveals that, for encoding, dense networks

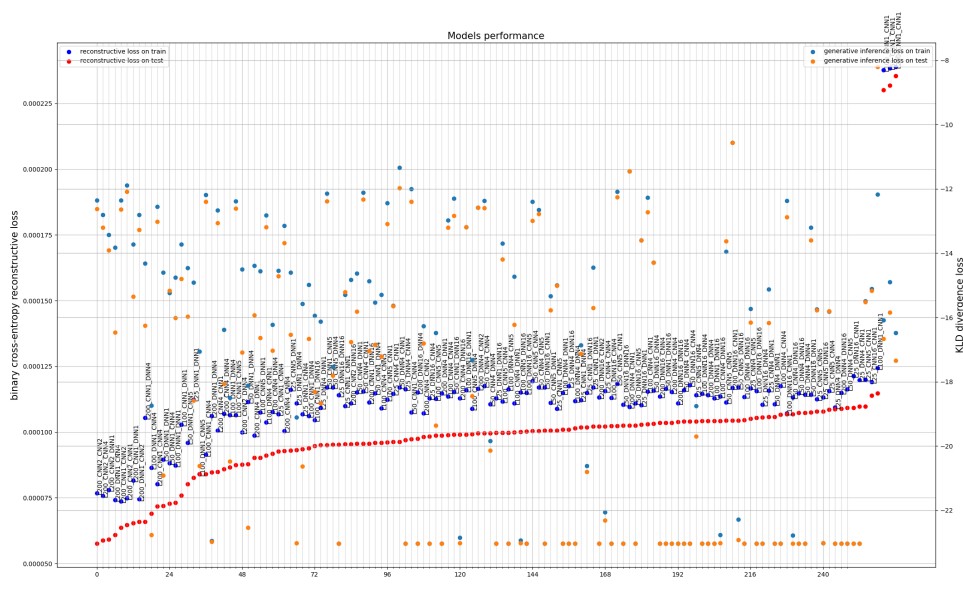

Figure 2: Performance of model combinations on the reconstructive loss. The right axis represents the generative inference loss (log scale). The left axis presents the reconstructionn error. Labels for each training follow the grammar L{latent space size}_{Encode architecture}{number of layers}_{Decoder architecture}{nnumber of layers}.

with only one layer generally outperform other configurations. Shallow architectures, particularly convolutional networks with a single block, also perform well. Interestingly, this trend holds for dense networks in decoding tasks, while convolutional networks tend to benefit from multiple convolutional blocks.

Further analysis of performance by architecture type (Figure 5) highlights the distinct advantages of convolutional layers in decoding tasks and the nuanced interplay between architecture choices and performance metrics.

### 4.3 LATENT SPACE COMPRESSION

The latent spaces generated by models with varying compression levels were evaluated for the top 25% (Figure 6) and top 50% (Figure 7) of models. Higher compression levels degrade the quality of the latent space, though they still manage to achieve separable representations at a 50% compression factor. However, models in the top 50% category, identified as having lower reconstruction capacities, generally struggle to find meaningful latent space projections.

## 5 CONCLUSION

This study highlights several key insights into the dynamics of encoder and decoder architectures in Variational Autoencoders (VAEs). A non-zero Kullback-Leibler Divergence (KLD) loss was found to be generally beneficial, as it contributes to maintaining meaningful latent representations. For encoding tasks, small and flexible networks performed better, likely due to the absence of structural processing bias, which allows for more adaptive learning of latent features. In contrast, decoding tasks benefited from architectures with structural processing capabilities, such as convolutional networks (CNNs) with multiple layers, which effectively leveraged the inherent spatial hierarchies in the data. Furthermore, powerful CNNs did not negatively impact encoding performance, suggesting that the encoder's capacity does not interfere with the decoder's ability to reconstruct data. Finally,

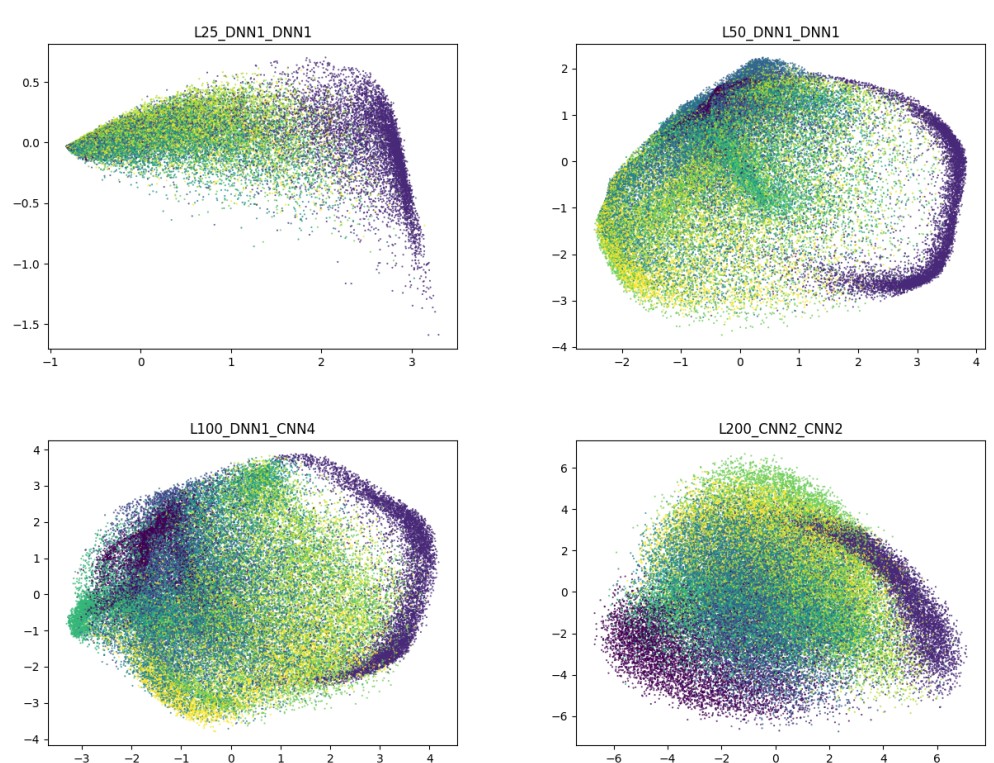

Figure 3: Top 25% performance of model combinations on the reconstructive loss. The right axis represents the generative inference loss (log scale).

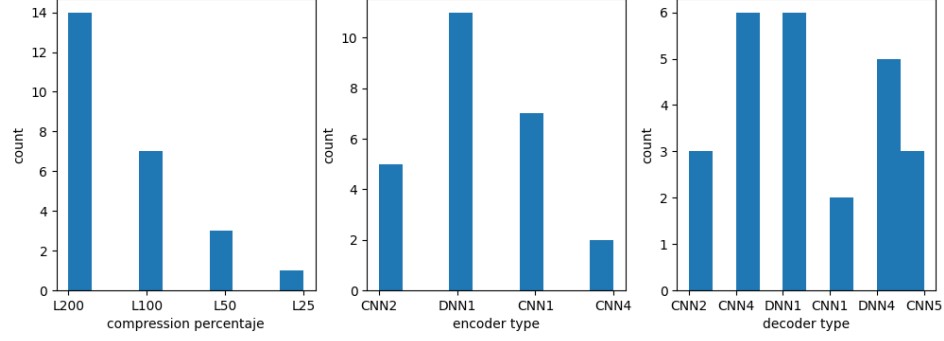

Figure 4: Left: Count of top-performing models by compression size. Center: Count of top-performing models by encoding architecture type. Right: Count of top-performing models by decoding architecture type.

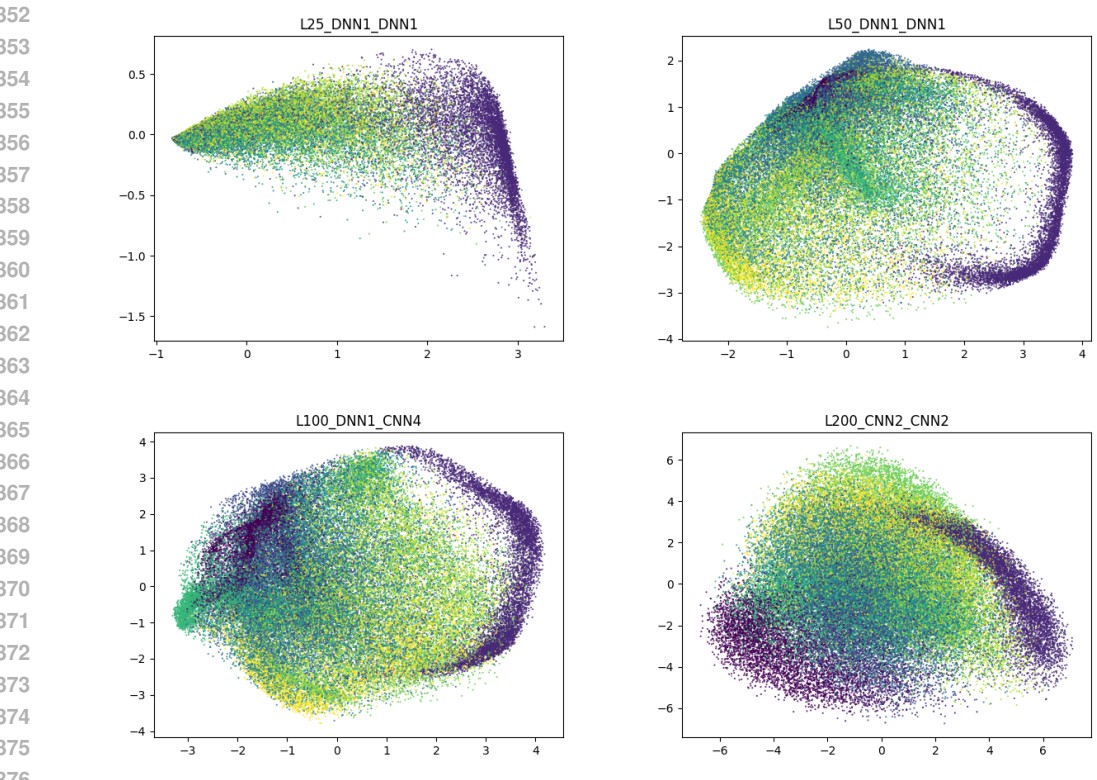

Figure 5: Top: Count of top-performing encoders by architecture type. Bottom: Count of top-performing decoders by architecture type.

Figure 6: Latent space projections for the top 25% of models.

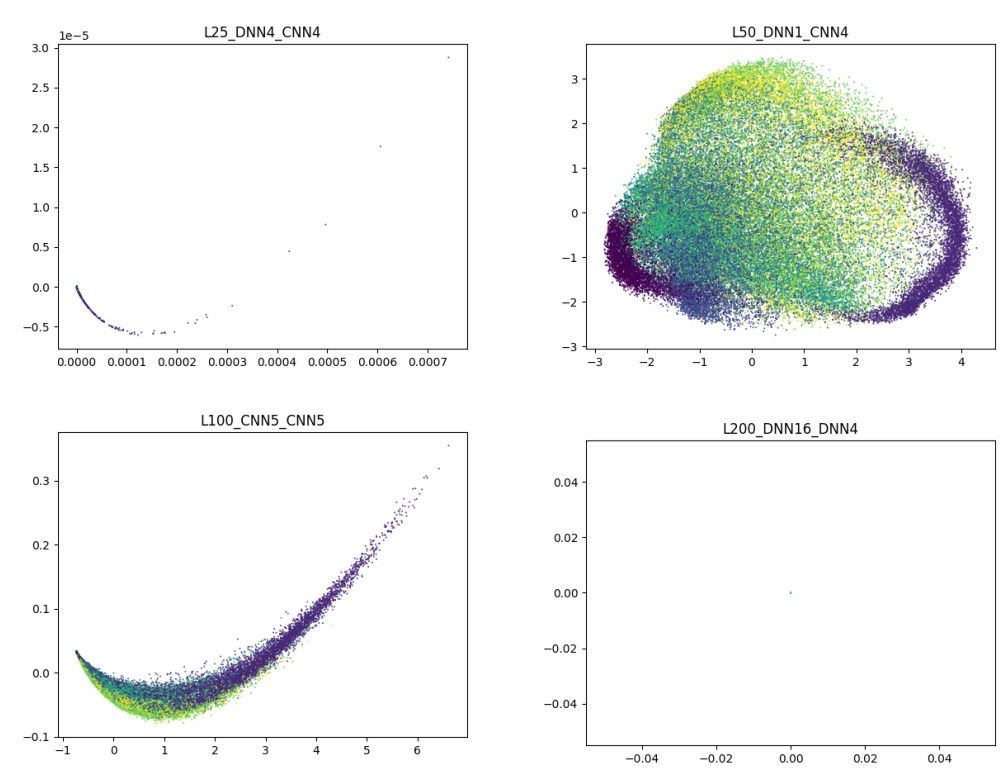

Figure 7: Latent space projections for the top 50% of models.

data compression proved challenging for multilayer perceptrons (MLPs), which struggled to effectively handle compact latent representations compared to other architectures.

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

## A  APPENDIX

You may include other additional sections here.

