# OpenReview forum: "When Encoders Should Stay Simple: An Empirical Analysis of Architectures for Variational Autoencoders"
_ICLR.cc/2026/Conference — Submitted to ICLR 2026_

### Official Review · Reviewer_c5Ym · 2025-10-29

**Soundness:** 1
**Presentation:** 1
**Contribution:** 1
**Rating:** 0
**Confidence:** 4

**Summary:**

In this work, the authors conduct an ablation study to investigate how different encoder–decoder architecture configurations affect model performance in terms of compression and reconstruction capabilities. They evaluate both the KL divergence between the variational posterior and the prior, as well as the reconstruction loss, across various architectural setups and latent space dimensionalities. The experiments are performed on the MNIST dataset, using architectures limited to dense and convolutional neural networks.

**Strengths:**

I believe that conducting a comprehensive study on how architectural choices influence the performance of VAEs, attempting to isolate the effect of architecture from other design factors (such as flexible priors, flexible posteriors, or tighter lower bounds) may be valuable.

**Weaknesses:**

Overall, I think the paper and its analysis lack sufficient technical depth and connection to the current state of the art. Moreover, the experimental design is quite limited. As a result, the paper fails to provide meaningful insights into its primary objective: understanding how architectural choices influence the performance of these models.

-	**Motivation.** I believe the motivation for this study is not clearly articulated. After reading the introduction, it remains unclear why the authors chose to focus on this particular family of generative models (for instance, they acknowledge that models such as GANs outperform VAEs in terms of generative quality, yet they do not explain why VAEs are still worth investigating).
-	**Connection to the current state of the art.** I think the paper does not adequately position itself within the current state of the art. In the introduction, the authors overlook significant developments in generative modeling since the introduction of VAEs and GANs over a decade ago. This includes advances within these model families as well as alternative approaches such as DDPMs and flow-based models. The paper does not discuss why investigating vanilla VAEs remains relevant in light of these developments. Furthermore, the discussion of VAE limitations is superficial, and the authors overlook existing work that comprehensively analyzes and addresses these issues, some of them proposing changes in the neural networks (see [1-6] for some examples), that could enrich the analysis of this paper. As a result, the paper fails to properly assess the state of the art, identify a clear research gap, or connect to prior analyses of how design choices influence model performance. The related work discussion is limited to only three articles, which is insufficient to capture the broader literature.
-	**Lack of technical depth.** The analysis of the state of the art, the related work, the motivation to study the effect of the architecture and the discussion of the experimental results remains at a very high level, and does not discuss the internal model dynamics that may explain the behavior observed in the experiments. Consequently, the discussion offers limited insight into how architectural choices influence model performance, leading to a rather weak overall contribution.
-	**Experimental design.** I think that the experimental setup is overly simplistic considering the significant advancements in VAE architectures (e.g., flow-based encoders, autoregressive decoders, hierarchical models). Moreover, since the experiments are conducted solely on the MNIST dataset (one of the simplest benchmarks) the findings provide limited insight into real-world or more complex scenarios. Additionally, the authors do not specify the assumed variational family or likelihood used in their experiments, which further weakens the reproducibility and interpretability of the results. The evaluation could be strengthened by including qualitative results for both generation and reconstruction, as well as standard quantitative metrics commonly used to assess VAE performance, such as PSNR for reconstruction quality and FID for generative quality.

REFERENCES:

[1] Lucas, J., Tucker, G., Grosse, R. B., & Norouzi, M. (2019). Don't Blame the ELBO! A linear VAE Perspective on Posterior Collapse. Advances in Neural Information Processing Systems, 32.

[2] Dai, B., Wang, Z., & Wipf, D. (2020, November). The Usual Suspects? Reassessing Blame for VAE Posterior Collapse. In International Conference on Machine Learning (pp. 2313-2322). PMLR.

[3] Dieng, A. B., Kim, Y., Rush, A. M., & Blei, D. M. (2019, April). Avoiding Latent Variable Collapse with Generative Skip Models. In The 22nd International Conference on Artificial Intelligence and Statistics (pp. 2397-2405). PMLR.

[4] Van Den Berg, R., Hasenclever, L., Tomczak, J. M., & Welling, M. (2018, January). Sylvester Normalizing Flows for Variational Inference. In 34th Conference on Uncertainty in Artificial Intelligence 2018, UAI 2018 (pp. 393-402). Association For Uncertainty in Artificial Intelligence (AUAI).

[5] Tomczak, J. M., & Welling, M. (2017). Improving Variational Autoencoders using Householder Flow, 2016. In Advances in Neural Information Processing Systems, Workshop on Bayesian Deep Learning (Vol. 1, No. 5).

[6] Gulrajani, I., Kumar, K., Ahmed, F., Taiga, A. A., Visin, F., Vazquez, D., & Courville, A. (2017, February). PixelVAE: A Latent Variable Model for Natural Images. In International Conference on Learning Representations.

**Questions:**

-	Why do the authors find relevant investigating vanilla VAEs in light of the current advancements in generative models?
-	What specific gap in the current literature are the authors addressing? A more comprehensive literature review is needed, as prior works have already examined the limitations of vanilla VAEs and proposed new architectural approaches to address them, including comparisons to standard architectures (particularly relevant given that architecture is the main focus of this paper).
-	Results should be reproduced in more complex datasets (still common benchmarks), such as ImageNet, CIFAR, CelebA to mention some.
-	Authors should benchmark more architectures employed in the existing literature, such as flow-based encoders or autoregressive encoders/decoders
-	Additional experimental results should be included, both qualitative and quantitative, as noted previously.
-	How would the authors relate the observed effects with the training dynamics of the model? As I mentioned before, I think the current discussion on the experimental results offers little insight.

---

### Official Review · Reviewer_HNgB · 2025-10-30

**Soundness:** 1
**Presentation:** 1
**Contribution:** 1
**Rating:** 0
**Confidence:** 5

**Summary:**

The paper presents an empirical study of the effect of different sizes and types of architectures in the encoder and decoder of a VAE on the MNIST dataset.

**Strengths:**

None.

**Weaknesses:**

**Lack of Scientific Contribution:**
The paper does not provide any novel or meaningful insight to the community.
- To begin with, the experiment is conducted solely on the MNIST dataset, which makes the results insufficient to draw any definitive conclusions, particularly for an empirical study that lacks theoretical justification.
- Such empirical studies also require experimenting with modern VAE architectures such as NVAEs to be able to draw a meaningful conclusion.

**Poor Presentation:**
- I couldn't find a clear agenda or the core questions the experimental section aims to answer; neither any structured organization of the experiments.
- The paper's experimental presentation is difficult to understand. First of all, the figure labels are tiny and difficult to read. There are way too many network structures presented in a single figure; the results could be split up into multiple figures based on the size or the type of the network architecture, and the analysis could be done based on how different network types (MLP/CNN) affect performance and how the size of these networks affects them.

Even though the weaknesses are acknowledged, I find this work to be more of an empirical benchmarking effort rather than genuine scientific research. For instance, the phenomenon of latent space collapse is already well known and does not constitute a new scientific discovery. To make a meaningful scientific contribution, I recommend that the authors investigate the issue more deeply, providing convincing evidence of which structural factors specifically cause the collapse and offering insights that could motivate the design of novel architectures to address it.

**Questions:**

-

---

### Official Review · Reviewer_a8XR · 2025-10-31

**Soundness:** 1
**Presentation:** 2
**Contribution:** 1
**Rating:** 2
**Confidence:** 5

**Summary:**

This is an empirical paper that isolates architecture as the variable of interest in VAEs on MNIST. The takeaway is: ensure KL isn’t suppressed to zero (aka posterior collapse); and avoid overly aggressive compression (here defined as latent space dimension). The design is intentionally simple LeakyReLU blocks, or convolutional blocks, with plain ELBO training objective, and straightforward diagnostics (loss terms + PCA). The work aims to give practitioners an architectural rule-of-thumb for small-image VAEs based on experiments on MNIST.

**Strengths:**

Originality:
Most prior VAE work explores new losses (β-VAE, InfoVAE, FactorVAE, NVAE, etc.) or expressive posteriors (flows, VampPrior). This paper considers a useful but underexplore perspective: deliberately fix the probabilistic formulation and isolates architecture as the independent variable.\
The idea that powerful decoders can lead to posterior collapse or that encoder bottlenecks matter is well-known. The originality here lies more in systematic (at least for MNIST) empirical analysis, not in theory or new methods.

Quality:
The paper isolates architectural variables effectively (same loss, same dataset, changing encoder/decoder families).

Clarity:
The takeaway message of using simple encoders with structured decoders is communicated accessibly.

Significance:
The paper confirms heuristics of VAEs on small scale data.

**Weaknesses:**

1)	Single dataset (MNIST). The findings may not generalize to natural images or other modalities.
2)	Limited metrics. No FID/KID for sample quality, no bits-per-dim (NLL) estimates. Reliance on reconstruction + KL and PCA seems too thin.
3)	Ambiguity in terminology. “Generative inference loss” appears to denote the KL term; that phrasing is confusing and should be standardized (ELBO, KL, reconstruction).
4)	Under-specified training protocol. Crucial choices (optimizer, LR, batch size, number of epochs, weight decay, KL annealing, early stopping, seed control) aren’t reported here; the results could be highly sensitive to these choices.
5)	No parameter/FLOP matching. Architecture depth/width comparisons are not normalized for parameter count or compute. Gains could reflect capacity rather than inductive bias.
6)	Statistical reporting. No error bars/repeats across seeds, no significance tests. Illustratively, in line 206, the authors mention a weak correlation that is not quantified statistically. Or in line 256, the authors mention a separable representation, but this term is not properly defined and statistically quantified.
7)	Unclear figures. Some figures (e.g. 3) are not clear to me, e.g. what do the different colour represent.
8)	No contextualisation of previous results. It is unclear how the empirical results relate to previous theoretical work (e.g. [1-4]) and well-known practical tricks (e.g. [5-6])

Refs:
[1] Lucas, James, et al. "Don't blame the elbo! a linear vae perspective on posterior collapse." Advances in Neural Information Processing Systems 32 (2019).
[2] Alemi, Alexander, et al. "Fixing a broken ELBO." International conference on machine learning. PMLR, 2018.
[3] Burgess, Christopher P., et al. "Understanding disentangling in $\beta $-VAE." arXiv preprint arXiv:1804.03599 (2018).
[4] Dai, Bin, Ziyu Wang, and David Wipf. "The usual suspects? Reassessing blame for VAE posterior collapse." International conference on machine learning. PMLR, 2020.
[5] Dieng, Adji B., et al. "Avoiding latent variable collapse with generative skip models." The 22nd International Conference on Artificial Intelligence and Statistics. PMLR, 2019.
[6] Fu, Hao, et al. "Cyclical annealing schedule: A simple approach to mitigating kl vanishing." arXiv preprint arXiv:1903.10145(2019).

**Questions:**

Beyond addressing the weaknesses mentioned above: Ablate results with different
i)	Training choices (e.g. beta anealing)
ii)	Data sets beyond MNIST



The authors conclude that “smaller and flexible” encoders perform best. It would be helpful to clarify what “flexible” means in this context, since smaller architectures are typically less flexible in representational capacity.

---

### Official Review · Reviewer_S1Wx · 2025-11-01

**Soundness:** 2
**Presentation:** 2
**Contribution:** 1
**Rating:** 2
**Confidence:** 4

**Summary:**

This paper examines how encoder and decoder architectures affect the performance of Variational Autoencoders (VAEs) on MNIST. The authors vary the encoder and decoder types (dense vs. convolutional), depth, and latent dimension, and report reconstruction losses, generative (KL) losses, and PCA projections of the latent space. They find that shallow, dense encoders perform best; convolutional decoders with multiple blocks improve reconstruction; moderate compression retains separability; and non-zero KL divergence correlates with better representations.

**Strengths:**

- **Originality:** The paper takes a narrowly focused yet somewhat fresh approach by isolating the architectural components of VAEs, encoders, and decoders, without altering inference objectives or priors. While not particularly novel, this controlled study design provides a clean empirical setup to analyze how structural variations affect the quality of latent representations.

- **Quality:** The experimental workflow, though limited to MNIST, is systematic in exploring combinations of dense and convolutional layers across different latent dimensions. The inclusion of reconstruction and KL-divergence tracking, along with latent PCA visualization, demonstrates a reasonable level of methodological care within the scope of the work.

- **Clarity:** The core motivation and methodology are explained in straightforward terms, with figures that broadly convey the trends between architecture type and model performance. Despite some labeling inconsistencies, the message that simpler encoders and structured decoders perform better is communicated clearly.

- **Significance:** The findings, though dataset-specific, have potential practical relevance for researchers and practitioners selecting baseline VAE architectures. The empirical evidence that encoder simplicity and moderate latent regularization improve representation stability could inform future model design or serve as a reference point for larger-scale studies.

**Weaknesses:**

- **Limited empirical scope:**  The paper’s experiments are restricted to the MNIST dataset and rely primarily on qualitative visualizations, such as PCA projections and reconstruction plots. This narrow evaluation limits the strength and generalizability of the findings. To better align with its stated goals, the study should include multiple datasets (e.g., Fashion-MNIST, CIFAR-10) and report quantitative metrics to demonstrate whether architectural effects persist across data domains.

- **Lack of novelty and theoretical grounding:**  The core claims, that smaller dense encoders and multi-block CNN decoders perform better, are intuitive and have been observed in prior works such as NVAE and β-VAE. The paper does not provide a theoretical or mechanistic explanation for these effects. To strengthen its contribution, the authors could analyze how encoder complexity influences mutual information or the balance between reconstruction and regularization terms in the ELBO.

- **Capacity mismatch and insufficient experimental detail:**  The architectures compared differ in depth and parameter count, confounding performance differences. Without normalizing for model capacity or reporting parameter counts, conclusions about architecture are unreliable. Requesting the author(s) to add key training details: optimizer, learning rate, batch size, number of epochs, and random seeds.

- **Weak and undefined evaluation metrics:**  The term “generative inference loss” is not defined, and the study omits standard evaluation measures such as FID. These metrics are crucial for assessing reconstruction quality, sample fidelity, and latent structure. Including them would make the analysis more rigorous and comparable to existing literature.

- **Outdated literature context:** The discussion omits recent developments in hierarchical VAEs [a] (Duan et al., 2023) and latent diffusion models [b] (Rombach et al., 2022). Situating the work within this broader context could clarify its role as a baseline diagnostic study rather than a novel modeling approach, and would prevent it from appearing disconnected from modern generative modeling trends.

[a] Duan, Zhihao, et al. "Lossy image compression with quantized hierarchical vaes." Proceedings of the IEEE/CVF winter conference on applications of computer vision. 2023.

[b] Rombach, Robin, et al. "High-resolution image synthesis with latent diffusion models." Proceedings of the IEEE/CVF conference on computer vision and pattern recognition. 2022.

**Questions:**

1. **Experimental Results:**
   Could the authors provide full details of the training setup, including the optimizer, learning rate, batch size, number of epochs, random seeds, and any regularization schemes used? Reporting these would allow assessment of reproducibility.

2. **Presentation and Definitions:**
   What exactly is meant by the term “generative inference loss”? Can the authors define this loss formally in the paper and clarify how it differs from standard reconstruction or KL-divergence terms? Additionally, can the figures (e.g., Fig. 1) be relabeled with clear axes, units, and legends to improve interpretability?

3. **Architectural Fairness:**
   Were encoder and decoder architectures matched for parameter count and computational cost when comparing dense versus convolutional models? If not, could the authors normalize capacity or report total parameters to confirm that observed differences are not simply due to model size or regularization effects?

4. **Quantitative Evaluation:**
   Why were standard generative model metrics, such as FID, not included? Would adding these quantitative metrics change the interpretation of the results or reveal new trade-offs between reconstruction and latent structure?

5. **Theoretical Understanding:**
   Could the authors elaborate on why VAEs are believed to underperform GANs in producing sharp and high-quality representations? Is it primarily due to the Gaussian posterior assumption or the limitations of the ELBO objective? A clearer theoretical discussion of these structural limitations could help situate the findings within broader generative-model literature.

---

### Meta-Review · Area_Chair_eJoG · 2026-01-06

**Summary:**

All reviewers recommend rejection (scores: 0/0/2/2). The consensus is that the paper falls short of ICLR standards due to limited scope, low novelty, and lacking experimental rigor.

**Reviewer Concerns:**

No rebuttal.

**Reviewer Scores:**

No rebuttal has been posted, so no change.

---

### Decision · Program_Chairs · 2026-01-26

Reject